# Multi-Approach Characterization of Novel Pyrene-Degrading *Mycolicibacterium austroafricanum* Isolates Lacking *nid* Genes

**DOI:** 10.3390/microorganisms11061413

**Published:** 2023-05-27

**Authors:** Natalia Maria Silva, Camila Lopes Romagnoli, Caio Rafael do Nascimento Santiago, João Paulo Amorim de Lacerda, Sylvia Cardoso Leão, Luciano Antonio Digiampietri, Cristina Viana-Niero

**Affiliations:** 1Department of Microbiology, Immunology and Parasitology, Federal University of São Paulo, São Paulo 04023-901, Brazil; 2School of Arts, Sciences and Humanities, University of São Paulo, São Paulo 03828-000, Brazil; 3Laboratory of Chemistry and Manufactured Products, Institute of Technological Research, São Paulo 05508-901, Brazil

**Keywords:** genomic analysis, polycyclic aromatic hydrocarbon, gas chromatography/mass spectrometry

## Abstract

Polycyclic aromatic hydrocarbons (PAHs) are chemical compounds that are widespread in the environment, arising from the incomplete combustion of organic material, as well as from human activities involving petrol exploitation, petrochemical industrial waste, gas stations, and environmental disasters. PAHs of high molecular weight, such as pyrene, have carcinogenic and mutagenic effects and are considered pollutants. The microbial degradation of PAHs occurs through the action of multiple dioxygenase genes (*nid*), which are localized in genomic island denominate region A, and cytochrome P450 monooxygenases genes (*cyp*) dispersed in the bacterial genome. This study evaluated pyrene degradation by five isolates of *Mycolicibacterium austroafricanum* using 2,6-dichlorophenol indophenol (DCPIP assay), gas chromatography/mass spectrometry (CG/MS), and genomic analyses. Two isolates (MYC038 and MYC040) exhibited pyrene degradation indexes of 96% and 88%, respectively, over a seven-day incubation period. Interestingly, the genomic analyses showed that the isolates do not have *nid* genes, which are involved in PAH biodegradation, despite their ability to degrade pyrene, suggesting that degradation may occur due to the presence of *cyp*150 genes, or even genes that have not yet been described. To the best of our knowledge, this is the first report of isolates without *nid* genes demonstrating the ability to degrade pyrene.

## 1. Introduction

Polycyclic aromatic hydrocarbons (PAHs) are compounds found in the environment due to the incomplete combustion of organic matter, as well as anthropogenic activities such as exploration, and the refining and use of petroleum derivatives [1,2,3]. They have complex structures with low solubility in water and remain in the environment for long periods; PAHs are considered hazardous pollutants and represent environmental and human health concerns [4,5,6]. Sixteen PAHs have been described as priority environmental contaminants due to their carcinogenic and mutagenic effects [1,2,3]. Removing these environmental contaminants is of great concern, and several remediation technologies have been studied. Bioremediation is a safe and cost-effective option; various Gram-positive and Gram-negative bacteria, fungi, and algae with an ability to utilize PAHs have been characterized [7,8,9,10,11].

Environmental mycobacteria are ubiquitous microorganisms characterized by a specialized envelope that protect them from destruction in nature. Several mycobacterial strains isolated from different environments have been shown to have the ability to degrade PAHs. *Mycobacterium vanbaalenii* strain PYR-1 was the first of these taxa to be described in 1988 [12,13]; it was isolated from a petroleum-contaminated estuarine sediment. Since then, other environmental mycobacterial strains isolated from contaminated soils of diverse origins have been shown to degrade different PAHs, such as pyrene, anthracene, phenanthrene, and fluoranthene [14,15,16,17,18,19,20,21,22,23].

PAH degradation by mycobacteria begins with enzymes called dioxygenases, or RHO (ring-hydroxylating oxygenases), and/or monooxygenases (CYP), which add oxygen molecules to aromatic rings, leaving the compound unstable. Subsequently, aromatic ring cleavage reactions lead to the tricarboxylic acid (TCA) cycle, generating carbon dioxide and water [24,25,26,27]. The genome sequencing of several PAH-degrading mycobacterial strains, including PYR-1, revealed the presence of a genomic island called region A, containing genes encoding dioxygenases (*nid*A, *nid*B, *nid*A3, *nid*B3, and *nid*B2) involved in initiating PAH degradation [19,21,28,29]. This region is exclusive to pyrene-degrading strains [19]. In addition, *cyp* genes were found to be dispersed in the genomes [25,26].

In a previous project, we investigated the diversity of *Mycobacteriaceae* species in different water samples at the Zoological Park in São Paulo, Brazil. In this study, it was possible to isolate 380 isolates of *Mycobacterium*; five isolates were identified as *M. vanbaalenii/austroafricanum*.

Using phenotypic tests and gas chromatography/mass spectrometry (GC/MS), this study aimed to evaluate the ability of these five isolates to degrade pyrene. Pyrene is considered a typical high-molecular-weight PAH and a model for the initial evaluation of microbial capacity for PAH biodegradation [30,31,32]. In addition, genomic analyses were carried out to verify the presence of dioxygenase and monooxygenase genes in these bacteria. The results show that these isolates can degrade pyrene but lack the aforementioned *nid* genes and the entirety of region A.

## 2. Materials and Methods

### 2.1. Bacterial Strains, Media, and Cultivation Conditions

Strains MYC038, MYC040, MYC211, MYC221, and MYC223, identified as *M. austroafricanum*, were isolated from sewage and a lake at São Paulo Zoological Park Foundation in Brazil [33]. Bacteria were grown on solid Middlebrook 7H10 media supplemented with 10% OADC (oleic acid, albumin, dextrose, and catalase) (Becton Dickinson, Franklin Lakes, NJ, USA) at 30 °C for 5 days.

### 2.2. Qualitative Assays of Pyrene Degradation Using the 2,6-DCPIP Assay and Double-Layer Plates

The 2,6-DCPIP (2,6-dichlorophenol indophenol sodium salt, sigma) assay is based on the redox property of this compound. It is blue when oxidized, and when reduced, it turns colorless. The assay was performed on Bushnell–Haas (BH) mineral medium containing 50 mg/L of pyrene (Sigma), as described by Kubota et al. [34], with minor modifications, as described below [35]. The isolates were pre-cultured in 5 mL of Luria–Bertani (LB) broth medium at 30 °C for 5 days and at 160 rpm. After that, each culture was centrifuged at 4000× *g* for 5 min and washed twice with 0.9% sterile saline. Bacterial suspensions were adjusted to 1.0 (OD_660 nm_) in a BioPhotometer (Eppendorf, Hamburg, Germany) spectrophotometer, and 100 μL was transferred to a 1.5 mL sterile microtube containing 850 μL of BH–pyrene medium. Then, 50 μL of 2,6-DCPIP solution (375 μg/mL) was added, and the mixture was completely homogenized using a vortex. The assay was incubated at 30 °C on a rotary shaker at 150 rpm for 28 days in the absence of light, and observed daily. As negative controls for this assay, incubations were performed with: (1) BH medium and DCPIP and (2) BH–pyrene medium and DCPIP. The assay was considered positive for pyrene degradation when the color changed from blue to colorless. At seven-day intervals, aliquots were seeded on LB plates to assess the viability of the mycobacteria.

Double-layer plates were prepared with basal salt medium (BSM) with 1.5% agar containing no carbon source [17]. The overlayer was prepared with 4 mL BSM containing 1% agar and 400 mg/L of pyrene (Sigma) [18,36]. Bacterial suspensions were adjusted to 0.2 (OD_660 nm_) and spotted on double-layer plates. The sample was incubated at 30 °C for up to 28 days in the absence of light, and observed weekly [18,36]. Positive results were evidenced by the visualization of bacterial growth surrounded by a clear zone. The reference strain, *M. vanbaalenii* PYR-1 (DSMZ 7251), was used as a positive control for both assays.

### 2.3. Evaluation of Pyrene Degradation via GC/MS

The isolates were pre-cultured in 10 mL of Luria–Bertani (LB) medium at 30 °C at 150 rpm in a shaker for 5 days. Then, the cultures were centrifuged at 4900× *g* for 10 min, and the precipitates were washed twice with sterile Milli-Q water. After this step, bacterial suspensions with an optical density of 0.2 (OD_660 nm_) were prepared in Bushnell–Haas (BH) medium containing 10 mg/L of pyrene with or without 0.5% peptone [37,38]. As abiotic controls for this assay, incubations were performed with: (1) BH medium and pyrene and (2) BH medium, pyrene, and 0.5% peptone. The assay was incubated at 30 °C on a rotary shaker at 150 rpm for 7 days, and aliquots were removed at 0, 3, and 7 days. For each 10 mL aliquot, p-terphenyl-d14 or anthracene was added as a recovery standard at final concentrations of 40 µg/mL (Accustandard, New Heaven, CT, USA). PAH extraction was performed four times with dichloromethane (Honeywell, Charlotte, NC, USA) in a separatory funnel. The extracts were filtered through sodium sulfate to remove any water residue, collected in volumetric flasks, and the volume was made up to 100 mL with dichloromethane. A 1 mL aliquot of each extract was transferred to a CG/MS vial, and a phenanthrene-d10 injection standard was added at a final concentration of 1 µg/mL (Accustandard, New Heaven, CT, USA). Additionally, each aliquot was seeded on LB agar medium to assess the viability of the bacteria.

Analysis was performed on a QP-2010S gas chromatograph/mass spectrometer (Shimadzu Corp., Kyoto, Japan) using a 30 m × 0.25 mm × 0.25 µm DB-5MS column (Agilent Technologies, Santa Clara, CA, USA). The injector was set to 280 °C, and the injection was performed in splitless mode. Helium was used as the carrier gas at 1 mL/min, and the oven program was as follows: 60 °C initial temperature, increasing at 10 °C/min up to 300 °C, and maintaining this temperature for 17 min. The transfer line was set to 320 °C and the ion source was set at 230 °C. The acquisition was performed in SIM (selected ion monitoring) mode, monitoring one quantification ion and two qualification ions for each compound [20]. The calibration was performed with five solutions containing pyrene and the standards described above, ranging from concentrations of 0.1 µg/mL to 2.0 µg/mL. Quality control was performed using blank extractions on the culture media without pyrene. Analyses were performed in triplicate for all extracts, and the recovery standards were monitored.

### 2.4. Amplification of Dioxygenase (nid) and Cytochrome P450 (cyp) Genes via PCR

Seven genes that play a role in breaking down PAHs were examined: five dioxygenases (*nid*A, *nid*B, *nid*B2, *nid*B3, and *nid*A3) and two monooxygenases (*cyp*150 and *cyp*151) [25,26,39,40]. DNA was extracted via thermal lysis, prepared using a suspension of bacterial colonies in 300 μL of sterile Milli-Q water. Samples were incubated at 95 °C for 10 min; 50 ng was used for each PCR. All primers used in this study are listed in Appendix A, along with sequences, fragment sizes, and references. For each PCR, cycling conditions available in the references were used. The primers used for amplifying the *nid*B3 and *nid*B2 genes in this study were designed based on the *M. vanbaalenii* PYR-1 sequences (GenBank accession number: CP000511.1) using the OligoAnalyzer 3.1 program. PCR tests were performed with Invitrogen reagents at total volumes of 50 μL, which contained: 1X PCR Buffer, 200 μM of each DNTP, 0.4 mM of each primer, and 1.5 U Taq DNA Polymerase. Amplification of the *nid*B2 gene was performed using 2 mM MgCl_2_; the *nid*B3 gene was amplified with 1.5 mM MgCl_2_. Annealing temperatures were 68 °C for *nid*B2 and 60 °C for *nid*B3. All PCR tests were performed on a VeritiTM 96-Well Thermal Cycler (Applied Biosystems, Foster City, CA, USA). DNA from *M. vanbaalenii* PYR-1 (DSMZ 7251) was used as a positive amplification control; the negative control consisted of all reagents required to perform a PCR without any DNA sample.

### 2.5. Genome Sequencing, Assembly, and Annotation

The extraction of genomic DNA from the isolates was performed using the QIAamp DNA mini kit (Qiagen, Hilden, Germany) with the addition of a previous step of bacterial lysis. The bacterial mass was added to a tube containing 0.06 g of zirconia beads measuring 0.5 mm in diameter (BioSpec) with 100 µL of TE 1X, and then, vortexed vigorously for 5 min, incubated at 80 °C for 10 min, and centrifuged at 5000× *g* for 2 min. After these steps, the methodology recommended by the manufacturer was followed.

Genomes were sequenced using the Illumina/MiSeq Platform with paired readings. Libraries were prepared using the Illumina Nextera XT Library Prep Kit. Quality analysis of the sequences was performed using the Phred Quality Score algorithm in BioNumerics 7.63 software, adopting a cut-off point equal to 30, which is equivalent to 99.9% accuracy [41]. A script developed by the authors in Perl programming language was used for the removal of adaptors and primers (https://knowledge.illumina.com/library-preparation/general/library-preparation-general-reference_material-list/000001314, accessed on October 2019). The scaffolding process was performed using the CONTIGuator web server with default values for the parameters (minimal length of a contig to be accepted in the analysis: 1000 bp; minimal coverage of a contig to be accepted in the analysis: 20%; minimal length of a significant blast hit: 1100 bp; blast e-value threshold: e-20; minimum ratio for the best replicon estimation in case of a conflict: 1.5) [42]. *M. vanbaalenii* PYR-1 was used as the reference genome. Genome assembly was performed using SPAdes version 3.11.1 [43] with k-mer sizes equal to 21, 33, 55, 77, 99, and 127; error correction and careful options were activated. The automatic annotation of each genome was carried out using the NCBI Prokaryotic Genome Annotation Pipeline and the PATRIC 3.6.3 platform (Pathosystems Resource Integration Center), available at https://www.ncbi.nlm.nih.gov/genome/annotation_prok/ (MYC038, MYC040, MYC2111: 2 September 2021; MYC223: 5 September 2021 and MYC221: 23 February 2021) and https://www.patricbrc.org (MYC038: 23 January 2020, MYC040: 18 February 2020, MYC211: 21 February 2020, MYC221: 18 February 2020 and MYC223: 28 October 2021), respectively. The *M. vanbaalenii* PYR-1 genome sequence was used as a reference (GenBank accession number CP000511.1). The data were also recorded using the RASTt 2.0 classic program (Rapid Annotations using Subsystems Technology). The genomes have been deposited in GenBank and are available under accession numbers: CP082189, CP082190, CP082191, CP070380, and CP082302 (MYC038, MYC040, MYC211, MYC221, and MYC223, respectively).

### 2.6. Bioinformatic Analyses

Analyses of the average nucleotide identity (ANI) and genome–genome distance (GGD), and in silico analysis equivalent to DNA–DNA hybridization, were performed to determine the taxonomic positioning of bacteria. Genomes were analyzed using the ANI tools, available at https://github.com/widdowquinn/pyani (accessed on May 2021), and GGDC (Genome Distance Calculator 2.1), available at http://ggdc.dsmz.de/ggdc.php (accessed on May 2021). Results between taxa presenting either ANI > 95% or GGD > 70%, or both, were classified as belonging to the same species. Taxa showing ANI ≤ 97% and GGD ≤ 80% were classified as subspecies [44,45]. Genomes of the isolates in this study were compared with that of reference strain PYR-1 (GenBank accession number CP000511.1) using the Genome Analysis Tools module of the BioNumerics program (AppliedMaths, Sint-Martens-Latem, Belgium) v.7.63, in order to compare the nucleotide sequences of region A, which contained the dioxygenase genes and was described as specialized in the degradation of PAHs. Additionally, the genomes were compared using the Gene Tags Assessment by Comparative Genomics (GTACG) platform [46]. The amino acid sequences encoded by nid genes of *M. vanbaalenii* PYR-1 were used to query the genomes of isolates in this study with BLASTp and tBLASTn [47].

Comparative analysis of genomes based on protein sequence similarity was also performed using the RAST Server program SEED-viewer tool. For this analysis, genome sequences of two PAH-degrading isolates belonging to *Mycolicibacterium* sp. (*M*. sp. JLS and *M*. sp. MCS, accession numbers NC_009077.1 and NC_008146.1, respectively) and the PYR-1 strain were included; a non-degrading strain, *M. tuberculosis* H37Rv (accession number AL123456.3), was incorporated as a negative control.

## 3. Results

The five *M. austroafricanum* strains in this study were screened for PAH degradation via 2,6-DCPIP and double-layer plate assays using pyrene as a substrate. All isolates exhibited positive reactions in the 2,6-DCPIP assay within 15 days of incubation, while the reference strain PYR-1 showed a positive result within 10 days (Table 1). It was possible to observe that negative controls remained unchanged for up to 28 days of incubation. All strains remained viable throughout the study period. Three strains from the study (MYC038, MYC211, and MYC223) and the PYR-1 strain showed clear zones around spotted bacteria grown on double-layer plates (Figure 1).

To quantify the degradation of pyrene, the isolates were analyzed via CG/MS (Appendix A). Initially, the isolates were tested after growth in BH medium with pyrene, except for MYC040, which did not show any growth under this condition. None of the isolates demonstrated an ability to degrade pyrene in seven days (Figure 2A). Two isolates (MYC038 and MYC040) were chosen to evaluate the ability to degrade pyrene in the presence of 0.5% peptone, an agent with surfactant activity, in order to reduce the surface tension between the microorganism and the liquid medium. Under this condition, it was possible to observe that the isolates MYC038 and MYC040 demonstrated a 92% and 69% ability to degrade pyrene in three days, respectively, and a 96% and 88% ability to degrade pyrene in seven days, respectively (Figure 2B,C).

For further characterization of these isolates, the presence of dioxygenase and monooxygenase genes, described as initiators of the PAH biodegradation process, was evaluated via PCR. The *nid*A gene was assessed using two pairs of primers, one of which was degenerate (GP). Positive amplifications for the dioxygenases and *cyp*151 genes were detected only with DNA from the PYR-1 strain. In contrast, the *cyp*150 gene was amplified for all isolates in this study, as well as for PYR-1, as shown in Table 1. These results suggested that, despite their ability to degrade pyrene, these bacteria have no dioxygenase genes and might have other degradation pathways that have not yet been described.

The genomes of these isolates were sequenced for taxonomic confirmation and to search for genes related to pyrene degradation. The assembly revealed that the genomes had sizes of 6.03–6.28 Mb and 67.7–67.9% CG content (Table 2). For comparison, the reference strain *M. vanbaalenii* PYR-1 had a genome size of 6.5 Mb and 67.8% CG content. The GGD and ANI values of the isolates MYC038, MYC211, MYC221, and MYC223 in comparison to the genome of the reference strain PYR-1 confirmed their identification (Table 3). The isolate MYC040 exhibited ANI ≤ 97% and GGD ≤ 80% compared with *M. vanbaalenii* PYR-1, and was then classified as a subspecies.

Dioxygenase genes are found in region A of the PYR-1 strain, and have been described as related to PAH degradation. The genomes were subjected to comparative analysis with a focus on this region, starting at position 494,420 and ending at position 642,917 of the genome of strain PYR-1, highlighted in Figure 3. The results show that all isolates from this study did not have the entirety of region A, and the absence of dioxygenase genes was confirmed in this analysis.

Additionally, to evaluate the presence of region A in the genomes of these isolates, a comparative analysis of proteins was performed with the isolates of this study, genomes of the PYR-1 strain, two isolates described as PAH degraders (*M*. sp. JLS and *M*. sp. MCS), and *M. tuberculosis* H37Rv as a negative control. The results showed that region A, highlighted in red, was present in the PYR-1 strain and in the JLS and MCS isolates, but absent from the negative control and all isolates in this study (Figure 4). This absence was confirmed by the platforms GTACG, BLASTn, and BLASTp.

## 4. Discussion

This study used multiple approaches to perform the analysis of five isolates of *M. austroafricanum* acquired from a collection of environmental mycobacteria obtained from aquatic sources from a zoo in São Paulo, Brazil. The main focus was the evaluation of their capacity of PAH degradation, as previously observed for other isolates from this mycobacterial species [12,18,19,20,38,48].

The five *M. austroafricanum* isolates studied here were able to degrade pyrene using the 2,6-DCPIP colorimetric test, and three isolates exhibited halo formation in the double-layer plate assay. These qualitative tests have been used for the initial characterization of hydrocarbon-degrading microorganisms [12,18,20,37,49,50,51,52]. To the best of our knowledge, there have been no comparative studies on the sensitivity of these phenotypic tests for screening hydrocarbon-degrading bacteria. One hypothesis explaining the different results is the difference in the culture media composition and the characteristic of each test; one is performed in a liquid medium, whereas the other is performed in a solid medium.

Quantitative analysis of the microbial degradation of pyrene via GC/MS was evaluated with and without peptone, an agent that decreases surface tension; degradation was only observed in the presence of peptone. One factor affecting hydrocarbon degradation is the low bioavailability of these compounds to microorganisms. PAHs have low aqueous solubility and bioavailability; thus, biosurfactants promote the reduction in surface and interfacial tensions, increasing the bioavailability of hydrocarbons and, consequently, contributing to the degradation process of the compounds [4,53]. In fact, after the addition of peptone to the medium, pyrene degradation could be confirmed and quantified, including with isolate MYC040, which grew well in these conditions. Although several bacterial isolates had the ability to degrade pyrene as the only carbon source, the PYR-1 strain only demonstrated hydrocarbon mineralization in the presence of sorbitol [12,21,27,38,54,55,56]. Sorbitol could not be used here because the five isolates from this study did not grow in the presence of sorbitol.

Importantly, it was possible to observe the viability of the strains during the pyrene degradation experiments via cultivation on LB medium. In this study, we did not evaluate changes in CFU numbers as a form of transition of the cells into a viable but non-culturable (VBNC) state. This transition state has been reported for many bacterial genera, including *Mycobacterium*, as a response to different types of stress [57,58,59].

Whenever investigated, strains of the *Mycobacteriaceae* family with the ability to degrade hydrocarbons show the presence of genes encoding dioxygenases, especially *nid*AB and *nid*A3B3, that initiate the biodegradation process [1,17,19,20,60]. The results revealed that the isolates in this study lacked the dioxygenase genes involved in the degradation of PAHs, but retained the ability to degrade pyrene. We could not find data in the literature concerning isolates of this genus with the ability to degrade PAHs in the absence of these genes. On the other hand, it was demonstrated that mutants of *M. vanbaalenii* PYR-1 with the *nid*A gene disrupted by transposons had a reduced pyrene degradation capacity, although the ability was not totally eliminated, when compared with a wild individual, suggesting the existence of a non-preferential degradation pathway that is not encoded by the *nid* genes [39]. To the best of our knowledge, this is the first report of isolates lacking dioxygenase genes that are able to degrade pyrene, suggesting that degradation may occur due to the unique presence of *cyp*150 genes, or even by genes that have not yet been described.

In conclusion, this study presents new insights into the existence of alternative PAH catabolic pathways associated with bacteria. Transcriptomic and proteomic experiments using the isolates described here are currently ongoing, furthering our understanding of alternative catabolic pathways.

## Figures and Tables

**Figure 1 microorganisms-11-01413-f001:**
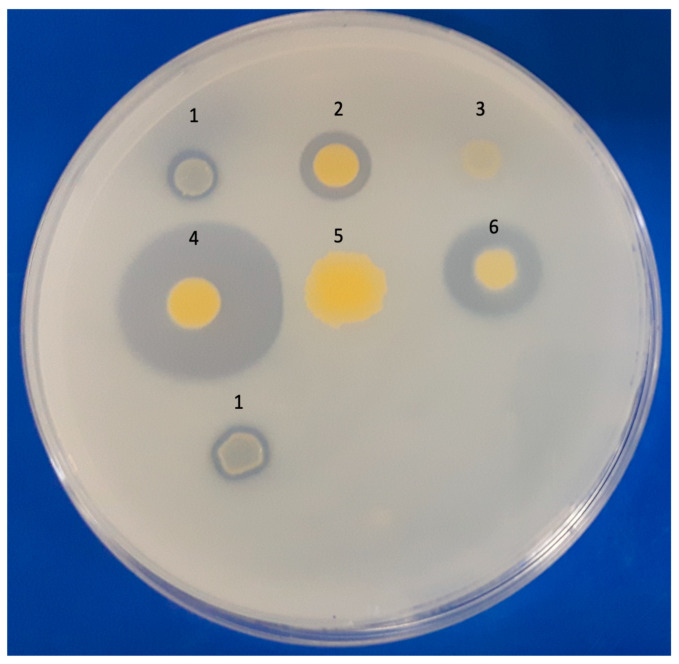
Phenotypic test of pyrene biodegradation on a double-layer plate. Positive control: (1) *M. vanbaalenii* PYR-1 DSMZ 7251. Isolates: (2) MYC038, (3) MYC040, (4) MYC211, (5) MYC221, and (6) MYC223.

**Figure 2 microorganisms-11-01413-f002:**
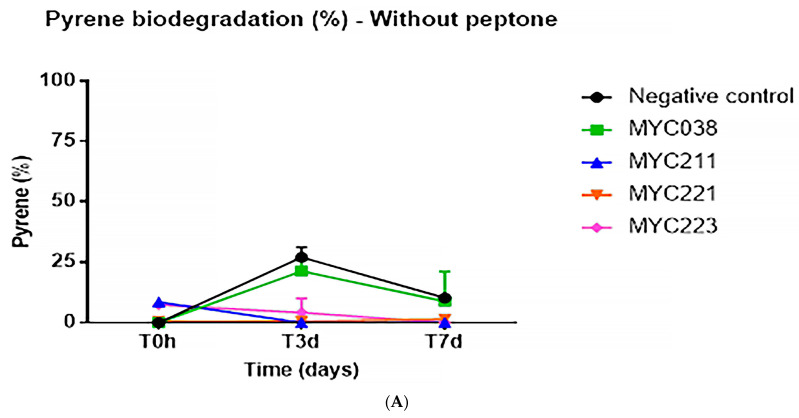
Pyrene biodegradation rates by MYC038, MYC040, MYC211, MYC221, and MYC223 isolates. (**A**) Pyrene biodegradation without peptone. (**B**) Pyrene biodegradation with 0.5% peptone. (**C**) CG/MS chromatograms showing biodegradation by MYC038 and MYC040 with peptone. T0h: black; T3d: blue; and T7d: red.

**Figure 3 microorganisms-11-01413-f003:**
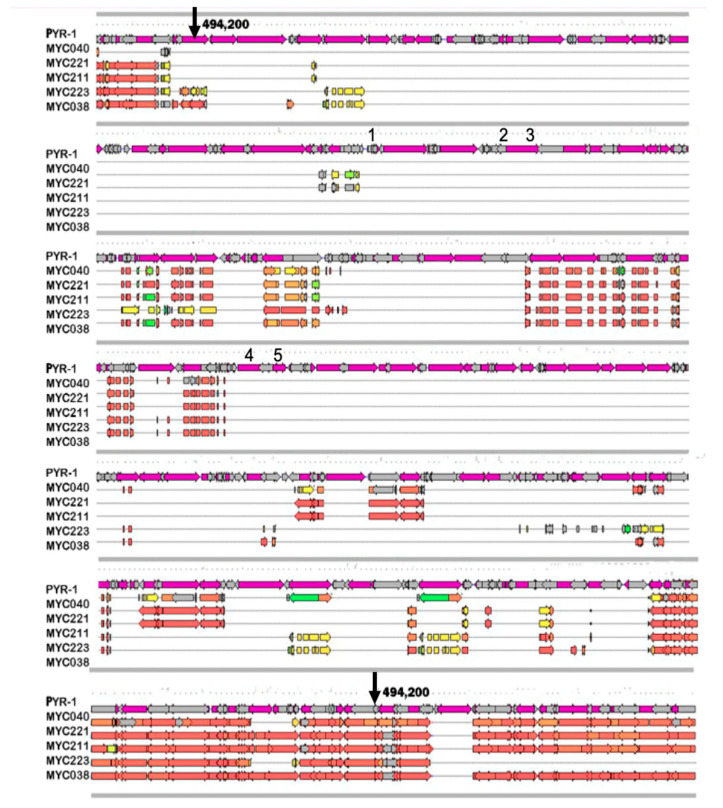
Schematic comparison of catabolic region A, responsible for PAH degradation, in *M. vanbaalenii* PYR-1 DSMZ 7251 and isolates MYC038, MYC040, MYC211, MYC221, and MYC223. In strain PYR-1, the start and end of region A are indicated by black arrows located at positions 494,420 and 642,917, respectively. This study analysed genes 1 to 5 (*nid*B2, *nid*B, *nid*A, *nid*A3, and *nid*B3). The ORFs found in the MYC isolates that matched the reference strain with 100% identity are colored red. ORFs with identity from 50% to 99% are marked in yellow or orange. ORFs with identity up to 50% are colored green, while ORFs that did not show identity are in gray.

**Figure 4 microorganisms-11-01413-f004:**
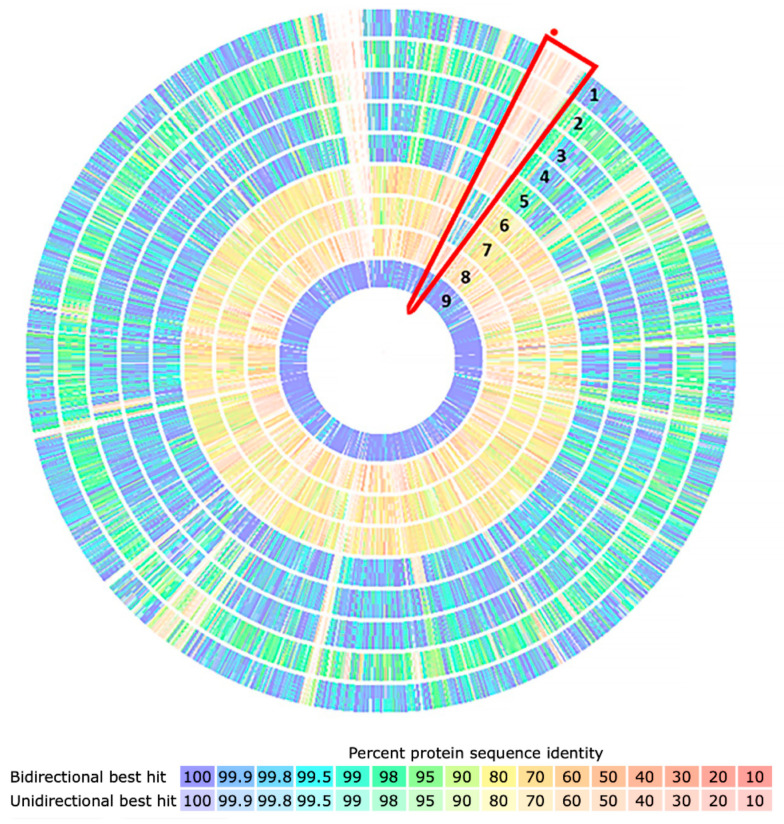
Comparative genomic analysis based on protein sequence identity. MYC038 (1), MYC040 (2), MYC211 (3), MYC221 (4), MYC223 (5), *M*. sp JLS (6), *M*. sp MCS (7), *M. tuberculosis* H37Rv (8), and *M. vanbaalenii* PYR-1 (9). The blue-to-red color scale indicates the percentage identity between the proteins, ranging from 100% to 10%. Highlighted in red is the A region present in the isolates with reports of PAH degradation and absence of the negative control (*M. tuberculosis* H37Rv) and isolates in this study.

**Table 1 microorganisms-11-01413-t001:** Phenotypic results of pyrene biodegradation via 2,6-DCPIP assay and PCR of *nid* and *cyp* genes. GP: degenerate primer used to assess the presence of the *nid*A gene.

Isolates	2,6-DCPIP	PCR							
		*nid*A	GP	*nid*B	*nid*B2	*nid*A3	*nid*B3	*cyp*150	*cyp*151
*M. vanbaalenii* PYR-1	+	+	+	+	+	+	+	+	+
MYC038	+	−	−	−	−	−	−	+	−
MYC040	+	−	−	−	−	−	−	+	−
MYC211	+	−	−	−	−	−	−	+	−
MYC221	+	−	−	−	−	−	−	+	−
MYC223	+	−	−	−	−	−	−	+	−

**Table 2 microorganisms-11-01413-t002:** Data from the sequencing, assembly, and annotation of isolates MYC038, MYC040, MYC211, MYC221, and MYC223.

General Features	MYC038	MYC040	MYC211	MYC221	MYC223
Genome size (MB)	6.26	6.03	6.18	6.28	6.17
Genome size (bp)	626,079	603,814	618,955	628,763	617,427
Paired-end reads (Illumina/MiSeq)	1,854,071	2,272,631	1,733,214	2,438,739	13,486,564
Number of contigs	99	113	95	121	79
N50 (bp)	149,987	118,372	147,758	139,998	133,202
Completeness (%)	100	100	100	100	100
G + C (%)	67.9	67.7	67.9	67.8	67.9
Number of protein-coding genes	5.883	5.628	5.751	5.853	5.755
Number of rRNA operons	2	2	2	2	4
Number of tRNAs	47	47	49	49	50
Sequence coverage (x)	70	91	68	96	510

**Table 3 microorganisms-11-01413-t003:** Comparative average nucleotide identity (ANI) and genome–genome distance (GGD) values for MYC038, MYC040, MYC211, MYC221, and MYC223 isolates and reference strain *M. vanbaalenii* PYR-1 DSMZ 7251.

	ANI %	GGD %
MYC038/*M. vanbaalenii* PYR-1	98.89	90.30
MYC040/*M. vanbaalenii* PYR-1	95.77	64.60
MYC211/*M. vanbaalenii* PYR-1	98.61	87.60
MYC221/*M. vanbaalenii* PYR-1	98.62	87.70
MYC223/*M. vanbaalenii* PYR-1	98.72	88.70

## Data Availability

All data are available in this manuscript.

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
