# Peer review of "Multi-Approach Characterization of Novel Pyrene-Degrading Mycolicibacterium austroafricanum Isolates Lacking nid Genes"

_microorganisms, 2023, doi:10.3390/microorganisms11061413_

Round 1
Reviewer 1 Report
In this study, the authors evaluated pyrene degradation by five isolates of Mycolicibacterium austroafricanum using 2,6-dichlorophenol indo-phenol (DCPIP assay), gas chromatography/mass spectrometry (CG/MS) and genomic analyses. The results showed that the pyrene-degrading strain did not possess nid genes, and the degradation may occur by the presence of cyp150 genes or even by genes not yet described. Overall, the topic is interesting and it could fit with this journal. However, the statement about the pyrene-degrading capability of the strains seems to be a little bit insufficient. I have the following specific comments for the improvement of this paper.
1. The Introduction was not well organized. In order to clearly demonstrate the biodegradation of PAHs, the living state of the strains should be investigated or discussed. As well known, most bacteria cannot be cultured on conventional bacteriological media due to entry into a viable but non-culturable (VBNC) state under harsh environmental conditions. In the VBNC state, bacteria exhibited low metabolic activity for long-term survival. Therefore, for comprehensive assessment of the microbial degradation of PAHs, the VBNC state should be considered. The latest publications which indicated the VBNC state of functional bacteria should be discussed in your paper: Xie et al, 2021, 87, e01110-21; Su et al., 2018, Letters in Applied Microbiology, 66, 277-283. Fida et al., 2017, Environmental Science & Technology, 51, 1570-1579.
2. In the Introduction section, the current research progress on the microbial degradation of pyrene should be added.
3. The novelty must be stated in the last paragraph of the Introduction.
4. For Materials and Methods, it would benefit from being condensed if possible. Methods might be condensed by elimination of excess detail about standard methods and by using references.
5. Line 69, 660 in OD660 needs subscript, check the whole manuscript.
6. Mycolicibacterium austroafricanum was given the full name when it first appeared, and it needs to be changed to M. austroafricanum later.
7. Avoid using first person (we, our, us), check the whole manuscript.
8. In section of “Qualitative assays of pyrene degradation by 2,6-DCPIP and double-layer plates”, the author claimed that the test was incubated at 30 ºC for up to 28 days. Why Figure 2A only presented the results for 7 days?
9. Results and discussion section - this is interesting however very descriptive - authors should more carefully justify their activity and experiments - why were those experiments performed and what was expected to be a major finding out of a particular experiment.
10. The author mentioned that GC/MS was used to detect the degradation metabolites of pyrene by the strain. Why only the GC/MS diagrams were shown in Fig.2C, and the corresponding metabolites and retention time were not listed? Please add relevant content.
11. Lines 254-260, the descriptions of region A in the genomes of these isolates are confusing, modifications should be carried out.
12. Figures (2, 3 and 4) are of poor quality and visual presentation. In particular, Fig. 2B is blank. Check them in the manuscript.
13. The manuscript needs to be carefully edited to improve the presentation and readability.
The manuscript needs to be carefully edited by a native English speaker to improve the presentation and readability.
Author Response
We appreciate and thank the reviewers' thorough review of this work and hope to have answered all questions. We carefully considered the reviewers’ comments to improve the manuscript, which was revised accordingly. The reviewers’ comments are in italics, and our responses (R) are in normal font. All changes in the manuscript are highlighted in red. In addition, the paper has undergone English language editing by MDPI, highlighted in blue.
- The Introduction was not well organized. In order to clearly demonstrate the biodegradation of PAHs, the living state of the strains should be investigated or discussed. As well known, most bacteria cannot be cultured on conventional bacteriological media due to entry into a viable but non-culturable (VBNC) state under harsh environmental conditions. In the VBNC state, bacteria exhibited low metabolic activity for long-term survival. Therefore, for a comprehensive assessment of the microbial degradation of PAHs, the VBNC state should be considered. The latest publications which indicated the VBNC state of functional bacteria should be discussed in your paper: Xie et al, 2021, 87, e01110-21; Su et al., 2018, Letters in Applied Microbiology, 66, 277-283. Fida et al., 2017, Environmental Science & Technology, 51, 1570-1579.
R: We agree with the reviewer. The introduction was rewritten, the methodology included the viability controls, and the suggested papers were cited in the discussion.
- In the Introduction section, the current research progress on the microbial degradation of pyrene should be added.
R: The introduction was rewritten, and a paragraph on pyrene degradation has been added to the introduction.
- The novelty must be stated in the last paragraph of the Introduction.
R: The last paragraph of the introduction was changed.
- For Materials and Methods, it would benefit from being condensed if possible. Methods might be condensed by elimination of excess detail about standard methods and by using references.
R: The methodology was extended and detailed at the request of the editors of the journal. We can change if necessary.
- Line 69, 660 in OD660 needs subscript, check the whole manuscript.
R: The text has been completely revised.
- Mycolicibacterium austroafricanum was given the full name when it first appeared, and it needs to be changed to M. austroafricanum later.
R: We agree with the reviewer. The text has been changed.
- Avoid using first person (we, our, us), check the whole manuscript.
R: We agree with the reviewer. The text has been changed.
- In section of “Qualitative assays of pyrene degradation by 2,6-DCPIP and double-layer plates”, the author claimed that the test was incubated at 30 ºC for up to 28 days. Why Figure 2A only presented the results for 7 days?
R: The DCPIP results are presented in Table 1, and those of the double layer in Figure 1. Tests were incubated for 28 days as described in the methodology and results section and according to other authors Zeng et al.,2010 (doi:10.1016/j.jhazmat.2010.07.085), Silva et al., 2019 (doi.org/10.1371/journal.pone.0215396).
Figure 2 shows the results of pyrene degradation obtained by GC/MS and this test was performed for 7 days as described in the methodology section and according to other authors ( Kim et al. 2007, Yang et al., 2021 and Yuan, et al., 2018) doi:10.1128/JB.01310-06, doi.org/10.1016/j.chemosphere.2020.127918, doi.org/10.1016/j.envpol.2018.09.001. In these papers, pyrene degradation has been evaluated between 3 and 14 days of incubation, and we follow them.
- Results and discussion section - this is interesting, however very descriptive - authors should more carefully justify their activity and experiments - why were those experiments performed and what was expected to be a major finding out of a particular experiment.
R: The discussion was rewritten as suggested by the reviewers.
- The author mentioned that GC/MS was used to detect the degradation metabolites of pyrene by the strain. Why only the GC/MS diagrams were shown in Fig.2C, and the corresponding metabolites and retention time were not listed? Please add relevant content.
R: the methodology used allows evaluating the degradation of pyrene but does not identify the metabolites formed. We did not find this information in the manuscript and changed some paragraphs, which may have led to a misunderstanding. In the results, line 228, is described that isolates were analyzed by CG/MS to quantify the degradation of pyrene (Supplementary Figures S1 and S2). In these figures are CG/MS chromatograms of negative controls and biodegradation by MYC038 and MYC040 with the retention time of compounds.
- Lines 254-260, the descriptions of region A in the genomes of these isolates are confusing, modifications should be carried out.
R: We changed the sentence and the description of region A.
- Figures (2, 3 and 4) are of poor quality and visual presentation. In particular, Fig. 2B is blank. Check them in the manuscript.
R: All figures were made available in 600 dpi according to the journal's rules. However, it is possible that they lost resolution when transforming them into a single document. Thus, all the figures were changed.
- The manuscript needs to be carefully edited to improve the presentation and readability.
R: The manuscript was revised and we hope to have made the necessary improvements. The English revision was carried out by MDPI.

Reviewer 2 Report
The results and organization of this article are relatively clear. In this study, different methods were used to identify the ability of Mycolicibacterium austroafricanum to degrade Polycyclic Aromatic Hydrocarbons (PAHs), and it was found for the first time that it possesses the ability to degrade PAHs even without the presence of NID genes. However, there are still some minor issues remaining.
Introduction:
The literature review on previous research on Mycolicibacterium austroafricanum and PAHs is insufficient. The introduction lacks sufficient information about the molecular background related to the main body of the article. Overall, the writing lacks logical coherence and depth of thought.
In lines 58-62, the study chose two polluted sites in Brazil, namely Cassia and Prudo, because they are considered to be representative of the oil-contaminated soil environments in Brazil.
Lines 219-221: If no bands are detected in the polymerase chain reaction (PCR), it does not necessarily mean that your gene of interest is not present. It is also possible that the primers do not amplify the gene in question due to low similarity between different species. Conversely, if a band of the expected size is detected, it is not necessarily the gene of interest, and sequencing validation is required after connecting to a T vector.
Regarding Figure 2A, why is MYC040 not included? Is it because it serves as the negative control? If not, what is the negative control?
Regarding Figure 2B, why am I not seeing any data presentation?
The discussion section is not a recap of your previous data, but a highly summarized analysis of the possible reasons for the data generated. It is not feasible to present repeated experimental data in the discussion section.
The present article did not analyze the reasons why the bacterium is capable of degrading despite lacking the NID gene, nor did it propose any possible molecular mechanisms. The article lacks prospective and sustainable perspectives.
The results and organization of this article are relatively clear. In this study, different methods were used to identify the ability of Mycolicibacterium austroafricanum to degrade Polycyclic Aromatic Hydrocarbons (PAHs), and it was found for the first time that it possesses the ability to degrade PAHs even without the presence of NID genes. However, there are still some minor issues remaining.
Introduction:
The literature review on previous research on Mycolicibacterium austroafricanum and PAHs is insufficient. The introduction lacks sufficient information about the molecular background related to the main body of the article. Overall, the writing lacks logical coherence and depth of thought.
In lines 58-62, the study chose two polluted sites in Brazil, namely Cassia and Prudo, because they are considered to be representative of the oil-contaminated soil environments in Brazil.
Lines 219-221: If no bands are detected in the polymerase chain reaction (PCR), it does not necessarily mean that your gene of interest is not present. It is also possible that the primers do not amplify the gene in question due to low similarity between different species. Conversely, if a band of the expected size is detected, it is not necessarily the gene of interest, and sequencing validation is required after connecting to a T vector.
Regarding Figure 2A, why is MYC040 not included? Is it because it serves as the negative control? If not, what is the negative control?
Regarding Figure 2B, why am I not seeing any data presentation?
The discussion section is not a recap of your previous data, but a highly summarized analysis of the possible reasons for the data generated. It is not feasible to present repeated experimental data in the discussion section.
The present article did not analyze the reasons why the bacterium is capable of degrading despite lacking the NID gene, nor did it propose any possible molecular mechanisms. The article lacks prospective and sustainable perspectives.
Author Response
We appreciate and thank the reviewers' thorough review of this work and hope to have answered all questions. We carefully considered the reviewers’ comments to improve the manuscript, which was revised accordingly. The reviewers’ comments are in italics, and our responses (R) are in normal font. All changes in the manuscript are highlighted in red. In addition, the paper has undergone English language editing by MDPI, highlighted in blue.
- The literature review on previous research on Mycolicibacterium austroafricanum and PAHs is insufficient. The introduction lacks sufficient information about the molecular background related to the main body of the article. Overall, the writing lacks logical coherence and depth of thought.
R: The introduction has been rewritten with more information, including HPA degradation by the PYR-1 strain.
- In lines 58-62, the study chose two polluted sites in Brazil, namely Cassia and Prudo, because they are considered to be representative of the oil-contaminated soil environments in Brazil.
R: The bacteria were isolated in a previous work that aimed to evaluate the biodiversity of mycobacteria in environments at the São Paulo Zoo. The bacteria analyzed here were isolated from sewage and lake water as described in line 76 (Romagnoli et al., 2020), doi.org/10.1371/jounal.pone.0227759.
- Lines 219-221: If no bands are detected in the polymerase chain reaction (PCR), it does not necessarily mean that your gene of interest is not present. It is also possible that the primers do not amplify the gene in question due to low similarity between different species. Conversely, if a band of the expected size is detected, it is not necessarily the gene of interest, and sequencing validation is required after connecting to a T vector.
R: The paragraph was rewritten, line 252, “For further characterization of these isolates, the presence of dioxygenases and monooxygenases genes, described as initiators of the PAHs biodegradation process, was evaluated by PCR. The nidA gene was assessed using two pairs of primers, one of them degenerate (GP). Positive amplifications for the dioxygenases and cyp151 genes were detected only with DNA from the PYR-1 strain. In contrast, the cyp150 gene was amplified for all isolates in this study and also for PYR-1, Table 1”.
Initially, we considered the hypothesis that there was no amplification of the genes due to alteration in the annealing site of the primers and, therefore, we used two sets to amplify the nidA gene, the GP primer being degenerate and even so, it did not generate amplification. Furthermore, genomic analyzes also did not locate the nid and cyp 151 genes in the genomes of the isolates.
- Regarding Figure 2A, why is MYC040 not included? Is it because it serves as the negative control? If not, what is the negative control?
R: MYC40 is not represented in Figure 2A because it does not grow in BH-pyrene medium without peptone (just as PYR does not grow only in pyrene); this information is shown in line 230. Figure 2B shows the results of pyrene degradation by MYC038 and MYC040 obtained in BH-pyrene medium plus 0.5% peptone. Negative controls were: 1) BH medium and pyrene and 2) BH medium, pyrene and 0.5% peptone, line 110.
- Regarding Figure 2B, why am I not seeing any data presentation?
R: The figure was again inserted in the text because there must have been some problem when transforming it into a single document.
- The discussion section is not a recap of your previous data, but a highly summarized analysis of the possible reasons for the data generated. It is not feasible to present repeated experimental data in the discussion section.
R: The discussion has been rewritten.
- The present article did not analyze the reasons why the bacterium is capable of degrading despite lacking the NID gene, nor did it propose any possible molecular mechanisms. The article lacks prospective and sustainable perspectives.
R: The discussion was rewritten. “To our knowledge, this is the first report of isolates able to degrade pyrene lacking dioxygenase genes, suggesting that degradation may occur by the unique presence of cyp150 genes or even by genes not yet described. In conclusion, this work brings new insights into the existence of alternative PAH catabolic pathways by bacteria. Transcriptomic and proteomic experiments with the isolates described here are being carried out to promote an understanding of alternative catabolic pathways".

Reviewer 3 Report
This work is focused on studying the ability of bacteria to degrade PAHs, which is an urgent task due to the increasing level of environmental pollution. The authors have done important work to assess the ability of five isolates to degrade pyrene. The bioinformatic analysis was carried out well. Data concerning the genes involved in PAHs biodegradation are useful and arouse interest in connection with a possible new pathway of PAHs catabolism. However, the manuscript needs a major revision before publication. The text of the manuscript needs to be carefully checked for errors in the design and edited of English language. In addition, there are not a number of experiments that the authors themselves point out in the Discussion section.
Please consider my comments as listed below.
Comments
To avoid grammar and linguistic mistakes, minor level English language should be thoroughly checked. Please revise your paper accordingly since several language issue occurs on several spots in the paper.
Carefully check the text for typos: italics for Latin taxa designations, capital letters after the comma, absence of spaces between words, etc.
Please provide space between number and units. Please revise your paper accordingly since some issue occurs on several spots in the paper.
Why only isolates MYC038 and MYC040 were chosen to evaluate the ability to degrade pyrene in the presence of 0.5% peptone? At the same time, strain MYC040 showed no ability to degrade pyrene in a phenotypic test on the double-layer plates.
Lines 277-279 “It should be noted that the concentration of pyrene used in the double-layer test was higher than the colorimetric test, which may have generated apparently discordant results.” Is it possible to do a low pyrene test to check this assumption?
Lines 294-295 “Thus, it is possible that the analyzed isolates degrade pyrene without the presence of peptone if the test was performed for 28 days.” Is it possible to test within 28 days to check this assumption?
Minor comments
Indexes indicating authors' affiliations should be in upper case
Italics for Latin names of genera and species
Italics for names of genes
Line 197 Typo "...nyd and cyp gened" instead of "…nid and cyp…"
Table 1. What the abbreviation GP means?
Please format Table 1 as required by the journal
Line 191 Pls remove the reference to Table 1, as the content of the table does not agree with the text that refers to it. I recommend moving table 1 after paragraph Lines 216-223.
Missing Figure 2B.
The title of the figure should be on the same page as the figure itself.
Line 228 “…of isolates MYC038, MYC040, MYC211 and MYC221 in comparison…”, what about MYC223 strain?
Line 237-239 if this is the caption for Table 2, then it should be drawn up in accordance with the rules of the journal
Line 249-253 if this is a caption for Figure 3, then it should be formatted according to the rules of the journal
Please make the "Author Contributions" according to the rules of the journal:
Author Contributions: For research articles with several authors, a short paragraph specifying their individual contributions must be provided. The following statements should be used “Conceptualization, X.X. and Y.Y.; methodology, X.X.; software, X.X.; validation, X.X., Y.Y. and Z.Z.; formal analysis, X.X.; investigation, X.X.; resources, X.X.; data curation, X.X.; writ-ing—original draft preparation, X.X.; writing—review and editing, X.X.; visualization, X.X.; su-pervision, X.X.; project administration, X.X.; funding acquisition, Y.Y. All authors have read and agreed to the published version of the manuscript.” Please turn to the CRediT taxonomy for the term explanation. Authorship must be limited to those who have contributed substantially to the work reported.
The references part must be written according to the requirements of the journal, where the year of publication is put in its correct place and with the correct italics and bold font style.
To avoid grammar and linguistic mistakes, minor level English language should be thoroughly checked. Please revise your paper accordingly since several language issue occurs on several spots in the paper.
Author Response
We appreciate and thank the reviewers' thorough review of this work and hope to have answered all questions. We carefully considered the reviewers’ comments to improve the manuscript, which was revised accordingly. The reviewers’ comments are in italics, and our responses (R) are in normal font. All changes in the manuscript are highlighted in red. In addition, the paper has undergone English language editing by MDPI, highlighted in blue.
- To avoid grammar and linguistic mistakes, minor level English language should be thoroughly checked. Please revise your paper accordingly since several language issue occurs on several spots in the paper.
R: The revision of the manuscript was carried out by the MDPI service, and we hope that it was enough.
- Carefully check the text for typos: italics for Latin taxa designations, capital letters after the comma, absence of spaces between words, etc.
R: The text was revised; probably, there was a problem when it was transformed into a single document.
- Please provide space between number and units. Please revise your paper accordingly since some issue occurs on several spots in the paper.
R: We apologize; there was certainly a problem when generating a single document containing text and figures.
- Why only isolates MYC038 and MYC040 were chosen to evaluate the ability to degrade pyrene in the presence of 0.5% peptone? At the same time, strain MYC040 showed no ability to degrade pyrene in a phenotypic test on the double-layer plates.
R: Due to budget limitations, we had to choose two isolates to evaluate the hypothesis of pyrene degradation occurring in the presence of a surfactant agent. Thus, the isolates were chosen in an exploratory way because the results of degradation by GC/MS were discordant with the phenotypic double layer and/or 2.6 DCPIP tests, which showed positive results. To the best of our knowledge, there are no comparative studies on the sensitivity of these phenotypic tests for screening hydrocarbon-degrading bacteria. A hypothesis to explain the different results would be the difference in the culture media composition and the characteristic of each test; one is performed in a liquid medium while the other is in a solid medium. This information was added in line 323.
- Lines 277-279 “It should be noted that the concentration of pyrene used in the double-layer test was higher than the colorimetric test, which may have generated apparently discordant results.” Is it possible to do a low pyrene test to check this assumption?
R: The double-layer test needs to present opacity, which is obtained by adding pyrene, to visualize a translucent halo around the bacterial colony as a positive result. In previous experiments, we tried to prepare the test with a lower concentration of pyrene as well as with other hydrocarbons to use as a screening, but we were unsuccessful.
- Lines 294-295 “Thus, it is possible that the analyzed isolates degrade pyrene without the presence of peptone if the test was performed for 28 days.” Is it possible to test within 28 days to check this assumption?
R: Yes, it is possible to carry out the test because we have already made the necessary standardizations. Unfortunately, at this moment, we do not have sufficient funds to carry out the experiment. As a result, we are writing a new project to request additional funding in order to continue our research.
Minor comments
- Indexes indicating authors' affiliations should be in upper case
R: the text has been revised.
- Italics for Latin names of genera and species
R: the text has been revised.
- Italics for names of genes
Line 197 Typo "...nyd and cyp gened" instead of "…nid and cyp…"
Table 1. What the abbreviation GP means?
R: In line 215 we include the abbreviation GP and also in the caption of Table 1.
Line 255: The nidA gene was evaluated using two pairs of primers, one of them degenerate (GP).
Table 1: GP: degenerate primer used to assess the presence of the nidA gene.
- Please format Table 1 as required by the journal
R: the table was revised according to journal rules.
- Line 191 Pls remove the reference to Table 1, as the content of the table does not agree with the text that refers to it. I recommend moving table 1 after paragraph Lines 216-223.
R: The reference has been removed from Table 1.
- Missing Figure 2B.
R: The figure was included again in the text.
- The title of the figure should be on the same page as the figure itself.
R: We agree; the review has been carried out.
- Line 228 “…of isolates MYC038, MYC040, MYC211 and MYC221 in comparison…”, what about MYC223 strain?
R: The sentence was changed, it was wrong, line 264.
- Line 237-239 if this is the caption for Table 2, then it should be drawn up in accordance with the rules of the journal.
R: The table was revised.
- Line 249-253 if this is a caption for Figure 3, then it should be formatted according to the rules of the journal
R: The figure was revised.
- Please make the "Author Contributions" according to the rules of the journal. The references part must be written according to the requirements of the journal, where the year of publication is put in its correct place and with the correct italics and boldfont style.
R: The author's contributions and references were changed according to the journal's rules as well as the English of the text was revised.
- To avoid grammar and linguistic mistakes, minor level English language should be thoroughly checked. Please revise your paper accordingly since several language issue occurs on several spots in the paper.
R: We agree and thank the reviewer, all text has been revised as suggested.

Round 2
Reviewer 1 Report
The revised manuscript has addressed all the comments by the reviewer. I recommend acceptance of the manuscript for publication.
Author Response
We are grateful for the careful review of the manuscript and suggestions that improved the understanding of our work.

Reviewer 3 Report
Thank you to the authors for working on correcting the comments.
I have only a few minor comments.
Minor comments
Line 80 The title “Qualitative assays of pyrene degradation using the 2,6-DCPIP assay on double-layer plates” may have missed the "and" between the two methods of analysis
Line 210 “All strains remain viable throughout the study period.” Please add a clarification (data not shown), since no data is given in the methods or in the results.
Line 299 “…and a degrader of M. tuberculosis H37Rv as a negative control…” Perhaps the term "degrader" is not needed here, or does this strain of M. tuberculosis also have some kind of degrading activity?
Author Response
We are grateful for the careful review of the manuscript and suggestions that improved the understanding of our work. The changes in lines 80, 210 and 299 were done.
